# Mediated Transformation of Tamarillo (*Solanum betaceum*) Callus Cell Suspension Cultures: A Novel Platform for Biotechnological Applications

**DOI:** 10.3390/plants14071028

**Published:** 2025-03-26

**Authors:** Ricardo Ferraz, Bruno Casimiro, Daniela Cordeiro, Jorge Canhoto, Sandra Correia

**Affiliations:** 1Centre for Functional Ecology, TERRA Associate Laboratory, Department of Life Sciences, University of Coimbra, Calçada Martim de Freitas, 3000-456 Coimbra, Portugal; rikayferr@hotmail.com (R.F.); casimiro19@gmail.com (B.C.); danielacordeiro@outlook.pt (D.C.); sandraimc@uc.pt (S.C.); 2Laqv Requimte, Departamento de Biologia, Faculdade de Ciências, Universidade do Porto, Rua do Campo Alegre s/n, 4169-007 Porto, Portugal; 3Faculty of Science, Agriculture & Engineering (SAgE), Newcastle University, Newcastle upon Tyne NE1 7RU, UK; 4Department of Life Sciences, University of Alcalá, 28805 Alcalá de Henares, Spain; 5InnovPlantProtect CoLAb, Estrada de Gil Vaz, 7350-478 Elvas, Portugal

**Keywords:** cell suspension cultures, GUS activity assay, in vivo monitoring, relative qPCR, Solanaceae

## Abstract

*Solanum betaceum* Cav. (tamarillo) has a strong biotechnological potential given the ease of obtaining cell lines from it that can be genetically transformed. However, genetic transformation of tamarillo cell suspension cultures has not yet been described. This study presents a simple method for *Agrobacterium*-mediated transformation of these cells and demonstrates the successful insertion of the β-glucuronidase gene (*gusA*) and the yellow fluorescent protein gene (*eyfp*) in their genome. For the success of this protocol, the selection of actively growing sub-cultured callus as explant and isolation of bacterial colonies with a cell density OD_600_ of 0.6–0.8 were key steps. Also, the inoculation of the callus in a bacteria liquid culture, the use of sonication, and the addition of antioxidants were essential. The transient expression of the *gusA* gene in tamarillo callus was confirmed and quantified, and no significant differences were observed between using LBA4404 or EHA105 strains. Finally, the insertion of the *eyfp* gene in the tamarillo genome enabled the in vivo confirmation of the transformation success. The present study showed that tamarillo cell suspension cultures can be genetically modified, opening the way for metabolite production in transformed cells and future scaling-up in bioreactors.

## 1. Introduction

Solanaceae, a cornerstone plant family, encompasses major food sources like tomato and potato as well as valuable model organisms in plant biotechnology, such as *Nicotiana tabacum* and related species [1]. Tamarillo (*Solanum betaceum* Cav.) is a solanaceous tree that has its origins in the South America Andean region [2]. The major economic interest of the tamarillo tree is related to the high nutritional and economic value of its fruits [3], presenting a global export value of 4.19 billion dollars in 2023 (www.tridge.com/intelligences/tamarillo/export accessed on 20 February 2025). This growing economical relevance is followed by significant biotechnological potential, particularly due to the easy use of different micropropagation protocols [4] and the establishment of cell cultures with different morphogenic competencies [5]. The non-competent callus, consisting of homogeneous, dedifferentiated cells with no capacity for organ regeneration [6], can be utilised for the production of valuable secondary metabolites [7] and biocatalysts with high demand for the industry. The availability of such plant cell cultures enables the enhanced production of specific metabolites with high interest, such as hydrolases in elicited tamarillo suspension cultures, which demonstrates the high potential of this species for scale-up processes [5,8].

Plant cell cultures as expression systems for recombinant proteins are inexpensive to grow and maintain. In addition, because plants are higher eukaryotes, they can carry out many of the post-translational modifications that occur in human cells. Plant cells are also intrinsically safe because they neither harbour human pathogens nor produce endotoxins [9]. Thus, recombinant peptides and proteins, such as antibodies, produced in plant cells present several advantages over production in microorganisms, such as large-scale production, safety, and post-translational modifications [10,11]. Furthermore, plant cells and tissue cultures have been studied and used to produce high-value molecules, such as bioactive secondary metabolites with commercial applications, namely shikonin, taxol, alkaloids, and berberine [12,13,14,15], and, more recently, the production of SARS-CoV-2 proteins by transient transformation of tobacco plants [16]. Recently, a study involving the analysis of tamarillo proteins to identify milk-clotting proteases isolated a serine protease with the potential for cheese making, alongside other proteases [17], emphasising the value of tamarillo active metabolites and the possible benefits of their production in cell suspension cultures.

In plant biotechnology, transient genetic expression presents a powerful alternative to permanent plant transformation, which is often time-consuming and susceptible to instability due to undesirable high-copy insertions of target genes. Transient expression enables rapid gene expression in short-term experiments, often used to verify transformation protocol efficacy [18,19]. This approach facilitates the production of recombinant proteins with minimal time for system modification and optimisation, making it a preferred method for inducing recombinant protein production. Among the available transient transformation methods, *Agrobacterium*-mediated transformation is especially efficient and productive [20].

Given the potential of tamarillo callus cell suspension cultures for producing valuable bioactive compounds, such as antioxidants and anti-inflammatory agents, this study aimed to develop the first optimised transient transformation protocol for tamarillo-induced callus lines using *Agrobacterium*. Accordingly, the transient transformation of tamarillo callus cells was confirmed by GUS activity assays and quantitative PCR, providing both qualitative and quantitative data, following the recent results obtained for tamarillo embryogenic callus by Cordeiro et al. [21]. However, in the present study and following Chu et al. [22], a second approach was also performed, involving the use of cell suspension cultures and infection in liquid medium, with successful transformation confirmed through EYFP expression visualisation. This additional approach represents a straightforward in vivo transformation method that could offer a useful platform for future production of molecules of interest using plant cells.

This protocol marks a significant advancement in tamarillo biotechnology, enabling the transformation of this species—particularly its callus suspension cells—as a novel platform for producing high-value molecules. Additionally, it facilitates scale-up, addressing the low-yield bottleneck often associated with plant cell systems.

## 2. Materials and Methods

### 2.1. Induction of Callus

The callus used in this study was obtained following the protocol used for somatic embryogenesis induction from young leaves of tamarillo [23]. The induction medium (TP) used was Murashige and Skoog [24] (MS) basal medium (Duchefa Biochimie, Haarlem, The Netherlands) supplemented with 9% (*w*/*v*) sucrose (Duchefa Biochimie) and 5 mg/L picloram (Sigma-Aldrich^®^, St. Louis, MO, USA), and 2.5 g/L Phytagel™ (Sigma-Aldrich^®^). The highly proliferative callus was sub-cultured monthly in test tubes with 12 mL of the same TP medium and maintained in the dark in a growth chamber at 24 ± 1 °C, as described in Casimiro et al. [5]. For the transformation procedure in liquid medium, 20 g of induced callus was grown in 100 mL of suspension liquid medium (TS) composed of half-strength MS basal medium with full vitamins, supplemented with 3% (*w*/*v*) sucrose, 5 mg/L picloram, 1 g/L casein hydrolysate (Sigma-Aldrich^®^), 50 mg/L glutamine (Sigma-Aldrich^®^), 20 mg/L cysteine (Sigma-Aldrich^®^) and ascorbic acid (Sigma-Aldrich^®^), and 2 mg/L glycine (Sigma-Aldrich^®^). The resulting cell suspension culture was maintained in 250 mL Erlenmeyer flasks in the dark in a growth chamber at 24 ± 1 °C with 120 rpm agitation for a week before the transformation step, ensuring the exponential phase had been reached. The pH of media was adjusted with 1 M KOH to 5.7 before autoclaving at 121 °C for 20 min.

### 2.2. Vectors, Constructs, and Agrobacterium Strains Used for Transformation

The genetic transformation assays were carried out with the following vectors: (1) p35SGUSINT [25], which contains the *ntpII* gene under the regulation of the *nos* promoter and terminator and the β-glucuronidase gene (*gusA*) coding region with an intron linked to the cauliflower mosaic virus 35S promoter (CaMV35S), located near the left border region, and (2) pMOG800 [22], which contains the selectable marker gene *nptII* driven by the *nos* promotor and the visual marker-enhanced yellow fluorescent protein gene (*eyfp*) interrupted by the IV2 plant intron and controlled by a 35S promoter. The successful incorporation of the p35SGUSINT plasmid into the *Agrobacterium tumefaciens* strains LBA4404 [26] and EHA105 [27] was carried out by Alves [28], and the resulting stock cultures were used in the following steps. The *A*. *tumefaciens* LBA4404 stock cultures containing the pMOG800 plasmid were kindly provided by Prof. Lorenzo Burgos Ortiz.

Bacteria were grown in Luria broth (LB) (Thermo Fisher Scientific^®^, Waltham, MA, USA) medium with 50 mg/L kanamycin (KAN) and 100 mg/L rifampicin (Duchefa Biochimie)-selective medium (SM). For the coculture procedure that used the *Agrobacterium* culture containing the p35SGUSINT plasmid, colonies were isolated from plates with selective medium and grown in 50 mL centrifuge tubes overnight with 5 mL of SM liquid medium at 28 °C and 220 rpm, until the culture attained an OD_600_ of 0.6–0.8. Then, centrifugation was performed at 4000× *g* for 8 min at room temperature. The supernatant was discarded, and the pellet was resuspended in 4 mL MS medium. Afterwards, tubes were centrifuged again at 4200× *g* for 8 min at room temperature. The supernatant was discarded, and the pellet was resuspended in 4 mL MS supplemented with 100 μM acetosyringone (Sigma-Aldrich^®^), 100 μM glutamine, 1% (*w*/*v*) polyvinylpyrrolidone (PVP) 40 (Sigma-Aldrich^®^), and 10% (*v*/*v*) coconut water (Sigma-Aldrich^®^), as reported by Cordeiro et al. [21]. The bacterial suspensions were incubated on a shaker (100 rpm) at 28 °C for 4 h.

As for the coculture procedure that used the bacteria culture containing the pMOG800 plasmid, colonies were isolated from plates with selective medium and grown in 50 mL centrifuge tubes overnight with 10 mL of SM liquid medium at 28 °C and 180 rpm. On the following day, the bacteria solution was diluted with SM liquid medium until the culture attained an OD_600_ of 0.1. The culture was then incubated for 10 h, again at 28 °C and 140 rpm, until achieving an OD_600_ of 0.6–0.8. Then, centrifugation was performed at 3800× *g* for 15 min at room temperature. The supernatant was discarded, and the pellet was resuspended in 30 mL MS medium supplemented with 100 μM acetosyringone, 100 μM glutamine, 1% (*w*/*v*) PVP 40, and 10% (*v*/*v*) coconut water. The bacterial suspensions were incubated on a shaker (140 rpm) at 25 °C for 5 h.

### 2.3. Cocultivation and Washing of the Callus: Infection in a Multiwell

In a multiwell plate, the bacterial suspension containing the p35SGUSINT plasmid prepared in the previous step was added to the callus in a ratio of 1 mg callus/2.5 mL bacteria solution. The plate was sealed and incubated at room temperature under vacuum for 10 min. Afterwards, the liquid medium with the bacteria was removed with a pipette, and the callus was transferred to a 10 cm plate with an agar base covered with a Whatman^®^ (Maidstone, UK) qualitative filter paper Grade 1, soaked with 1 mL of MS medium and 100 μM acetosyringone. The plates were sealed, and cells were maintained in the dark in a growth chamber at 24 ± 1 °C for three days.

After three days, the callus was washed under a filter vacuum with 50 mL of MS liquid medium with 500 mg/L each of carbenicillin (CARB) and cefotaxime (CEFO) (Duchefa Biochimie). Each mass was transferred to a plate with TP solid medium with 250 mg/L each of CARB and CEFO. Then, the plates were sealed and maintained in the dark in a growth chamber at 24 ± 1 °C for three days.

### 2.4. Cocultivation and Washing of the Callus: Infection in Liquid Medium

After being vacuum-filtered, 2 g of callus from the 7-day suspension culture were added to 10 mL of the bacterial suspension containing the pMOG800 plasmid prepared earlier, in a 25 mL Erlenmeyer flask. This coculture was incubated in the dark at 24 °C with shaking at 110 rpm for 12 min. Following this, each sample was sonicated for 5 min with 35 kHz using Sonorex RK 31 (Bandelin Electronic GmbH & Co. KG, Berlin, Germany) and then incubated again under the same conditions for another 12 min. The solution was then filtered using previously sterilised 100 µm sieves in a laminar flow chamber. The collected callus was divided into 500 mg portions, and each portion was transferred to a 10 cm plate containing an agar base covered with two sheets of Whatman^®^ qualitative filter paper Grade 1, soaked with 1 mL of TS liquid medium without picloram and supplemented with 100 µM acetosyringone. The plates were sealed and incubated in the dark in a growth chamber at 24 ± 1 °C for three days.

After the three-day incubation, the callus was incubated in 10 mL of full TS liquid medium supplemented with 500 mg/L of CEFO in 50 mL centrifuge tubes for 2 min and 30 s at 110 rpm. The callus was then vacuum-filtered and resuspended in 8 mL of TS liquid medium supplemented with 300 mg/L CEFO and 50 mg/L KAN in 50 mL Erlenmeyer flasks. This resuspension was incubated in the dark at 24 °C and 110 rpm for three days in an orbital shaker.

Following this, the cell suspensions were allowed to settle, and the supernatant was discarded in a laminar flow chamber. The deposited cells were transferred to Petri dishes containing TS medium with 300 mg/L CEFO, 50 mg/L KAN, and 2.5 g/L Phytagel™. The dishes were sealed and incubated in the dark at 24 °C for three weeks.

### 2.5. Selection of Transformed Callus

To select transformed cells, calluses from both transformation procedures were transferred to a TP solid medium. For the ones from the multiwell transformation, the medium was supplemented with 50 mg/L KAN for selection, 250 mg/L each of CARB and CEFO, and 250 mg/L of CEFO for the transformation in the liquid medium for bacterial removal. The plates were sealed and maintained in the dark in a growth chamber at 24 ± 1 °C for three weeks. This procedure was repeated every three weeks, following the selective scheme presented in Table 1, until the callus showed a yellowish colour with no evidence of cell necrosis. During selection, the cells were observed with the Leica MC170 HD camera (Leica, Wetzlar, Germany) using the LAS EZ software version 3.4 (Leica).

### 2.6. Histochemical GUS Assay

The GUS activity assay was performed according to Hu et al. [29] and Schauer et al. [30] with slight modifications. Callus masses transformed in the multiwell that were resistant to kanamycin and control samples were soaked in 500 µL of X-Gluc (5 mM K_3_Fe(CN)_6_, 5 mM K_4_Fe(CN)_6_, 1 µM Na·EDTA·2H_2_O, 32 mM NaH_2_PO_4_, 68 mM Na_2_HPO_4_, 1% (*v*/*v*) Triton-X, and 3 mg/mL X-Gluc (5-bromo-4-chloro-3-indolyl-β-D-glucuronide (Thermo Fisher Scientific^®^) prepared in N,N-dimethylformamide)) and incubated overnight at 37 °C in the dark. After washing with 70% (*v*/*v*) ethanol, cells were incubated in lactic acid/glycerol/PBS (phosphate-buffered saline) (1:1:3) for 4 h at room temperature in the dark and then observed with the SMZ800N stereomicroscope (Nikon, Tokyo, Japan), with the Leica MC170 HD camera (Leica), and the LAS EZ software version 3.4 (Leica).

### 2.7. PCR Confirmation of Transgenic Cells

Firstly, three 100 mg samples of kanamycin-resistant calluses transformed in the multiwell using either LBA4404 or EHA105 strains and two 100 mg control samples were macerated in liquid nitrogen, and the respective genomic DNA was extracted using the NucleoSpin™ Plant II kit (Macherey-Nagel, Düren, Germany). DNA concentration and quality were determined using the NanoDrop™ One/OneC Microvolume UV–vis spectrophotometer (Thermo Fisher Scientific^®^). For the polymerase chain reaction (PCR), 1 µL of genomic DNA from each sample was mixed with the forward and reverse primers (0.25 µM each) and 12.5 µL of NZYTaq II 2× Green Master Mix (NZYTech, Lisbon, Portugal) in a final volume of 25 µL. The forward (5′-CATGAAGATGCTTCG-3′) and reverse (5′-ATCCACGCCGTATTCGG-3′) primers were used to amplify a 636 bp fragment from the *gusA* gene. The PCR program started with an initial denaturing step at 95 °C for 5 min, followed by 27 cycles of a denaturing step at 95 °C for 1 min, an annealing step at 65 °C for 1 min, and an extension step at 72 °C for 1 min, and the process ended with a final extension step at 72 °C for 5 min. This reaction was performed in an Arktik™ Thermal Cycler (Thermo Fisher Scientific^®^). Electrophoresis separated the PCR products in an agarose gel (0.8% (*v*/*v*)) stained with GreenSafe Premium (NZYTech) (3% (*v*/*v*)). Apart from the DNA samples, a molecular weight marker (NZYDNA Ladder III, NZYTech) was loaded in the gel.

### 2.8. qPCR Relative Quantification of the gusA Gene Insertion

To quantify the insertion of the *gusA* gene in the genomic DNA of the cells transformed in the multiwell, a qPCR was conducted using 2.775 ng of genomic DNA extracted from cells resistant to kanamycin and control samples, 5 µL of NZYSpeedy qPCR Green Master Mix (2×) (NZYTech), and the primers mentioned above for the amplification of the *gusA* gene (0.4 µM each), with a final volume of 10 µL. The *ACTIN* gene was used as a reference gene, using the primers referred to in Cordeiro et al. [31]—primer forward 5′-CCATGTTCCCGGGTATTGCT-3′; primer reverse 5′-GTGCTGAGGGAAGCCAAGAT-3′. The qPCR program started with an initial denaturing step at 95 °C for 3 min, followed by 40 cycles of a denaturing step at 95 °C for 5 s, and an annealing and extension step at 60 °C for 25 s. This was followed by a melting curve step, with a temperature gradient from 65 °C to 95 °C with fluorescence readings acquired at 0.5 °C increments, to further confirm the primers’ specificity. The assay included non-template controls, and all reactions were run in two technical replicates. Reactions were performed in 96-well plates and run in a CFX96 Real-Time System (Bio-Rad^®^, Hercules, CA, USA). The relative expression was calculated according to the 2^−ΔΔCt^ method [32].

### 2.9. Detection of Transformed Cells by Fluorescence Microscopy

When the callus transformed in the liquid medium revealed yellowish colour masses with no evidence of cell necrosis, i.e., appearance of transformed masses, the transformed callus and control samples were observed under a Zeiss LSM 710 (Carl Zeiss AG, Oberkochen, Germany) confocal microscope with magnification of 40× and 0.99 AU of pinhole diameter. EYFP fluorescence was viewed by excitation with a 488 nm Argon laser, a detection wavelength of 516–617 nm, and a detector gain of 627 for the EYFP channel and 500 for brightfield. Brightness and contrast were adjusted using the Zeiss Zen 2 (Blue Edition, Carl Zeiss AG) software.

### 2.10. Statistical Analysis

All experiments were carried out using at least four biological replicates, and all results are displayed as the averages of the replicates, along with their standard deviation values. The analysis of the statistical significance of the data obtained in the qPCR relative quantification was performed in GraphPad Prism (version 8.0.0 for Windows, GraphPad Software, San Diego, CA, USA, www.graphpad.com). Firstly, the homogeneity of variances was accessed by Cochran’s C test. Then, as the variances obtained were not homogeneous, a Mann–Whitney U-test was performed to verify the statistical significance of the data.

## 3. Results

### 3.1. Transformation of Tamarillo Callus Cells

In the present section, the *Agrobacterium* strain that would contribute to a successful *Agrobacterium*-mediated transformation of tamarillo callus cells is addressed, following the protocol described by Cordeiro et al. [21], with the main results summarised in the workflow chart presented in Figure 1 and described in the following sub-sections.

For the transformation protocol, we used the p35SGUSINT vector incorporated in EHA105 and LBA4404 *Agrobacterium tumefaciens* strains (Figure 1(1)–(3)). The bacteria were resuspended in an MS inoculation medium before the cocultivation. To reduce the browning of the callus, the inoculation medium was supplemented with acetosyringone (100 µM), glutamine (100 µM), PVP 40 (1% *w*/*v*), and coconut water (10% *v*/*v*) (Figure 1(4)), as optimised in Cordeiro et al. [21].

The cocultivation with the bacterial suspension was made in a multiwell plate in a ratio of 1:2.5 (mg callus/µL bacteria suspension), which proved to be a successful procedure concerning the cell recovery. Also, the vacuum infiltration ensured that all cells were submerged and in contact with the bacterial solution (Figure 1(5)), enhancing the infiltration by the bacteria.

Concerning the callus incubation (Figure 1(6)), it was crucial to perform the process on Whatman^®^ Grade 1 qualitative filter paper soaked with MS liquid medium containing acetosyringone and placed in Petri dishes with agar. This procedure ensured that the cells would not desiccate. Following this, the bacterial liquid culture was removed.

The callus-washing steps (Figure 1(7)) were also determinant of the outcome of this protocol. To ensure that most bacteria were removed, the cells were washed under vacuum with MS liquid medium supplemented with CARB (500 mg/L) and CEFO (500 mg/L).

### 3.2. Selection of Transformed Callus Cells

As Figure 2A shows, the callus cells at the final selection procedure showed limited growth and whitish areas (Figure 2(A2,A4)) compared with the beginning of the transformation procedure (Figure 2(A1,A3)) for both strains. The dark areas of the calluses were discarded at the end of each transfer, particularly in the fourth (12 weeks). Also, the bacteria growth observed in the first three weeks was not observed in the subsequent weeks.

### 3.3. Transformation Confirmation by β-Glucuronidase (GUS) Activity Analysis

The results of the GUS activity assay performed to confirm the genetic transformation process are illustrated in Figure 2B. As shown in Figure 2(B1), GUS staining was not found in control samples. On the other hand, all samples used in the transformation procedure showed the blue staining typical of the GUS activity (Figure 2(B2,B3)) without visible differences between each strain.

### 3.4. Transformation Confirmation and Efficiency Evaluation by PCR-Based Approaches

As shown in Figure 2(C1), all the treated samples amplified a 636 bp fragment corresponding to the *gusA* gene, confirming that the transformation procedure succeeded.

To quantify the insertion of the *gusA* gene in the genomic DNA of the callus cells, a qPCR was performed using cells resistant to KAN and control samples.

Figure 2(C2) shows the relative fold change. Despite no significant differences between the results of both strains, it can be concluded that the *gusA* gene was inserted into the genome of the cells used in the transformation protocol. This result confirms the observations made with the β-glucuronidase (GUS) activity analysis presented in Figure 2B.

### 3.5. Optimisation of the Protocol Using Liquid Cultures

In a second approach, building on the previous results and following the methodology described by Chu et al. [22], the tamarillo callus cells’ genetic transformation protocol was optimised to enable the transformation of cell suspension cultures through an infection step in a liquid medium. This method allows for the in vivo confirmation of transformation success without impairing any cells, as evidenced by the visualisation of fluorescence emitted from transformed cells containing the *eyfp* gene.

The liquid coculture was incubated twice with sonication in between, as sonication has been shown to enhance transformation efficiency [22]. A key optimisation step was the washing process since after it, no apparent bacterial presence was observed.

The selection procedure was conducted according to Table 1 (see Section 2). Initially, the antibiotic supplementation was intensified with higher concentrations of CEFO, which were gradually decreased and replaced with CARB in subsequent stages. This approach aimed to eliminate bacteria present in the cultures during the first weeks while preventing the slowdown of callus cells’ growth without completely removing the antibiotic.

For confirmation of transformation efficiency, Figure 3 illustrates the differences between control and transformed samples. Unlike the scattered and pale autofluorescence emitted by control samples (Figure 3A–F), the transformed ones (Figure 3G–L) exhibited strong *eyfp* gene expression around the nucleus of the cells, likely in the endoplasmic reticulum, confirming the success of the transformation procedure. The cells appeared healthy, maintaining the integrity of the cytoplasm and cell membrane.

## 4. Discussion

Tamarillo is valued not only as a food source but also for its high potential as a biotechnological tool, particularly in the production of molecules of interest through cell suspension cultures [5]. Previous studies on tamarillo have demonstrated successful *Agrobacterium*-mediated genetic transformations, with regeneration achieved through organogenesis [33] and somatic embryogenesis [21,34]. However, a comprehensive protocol for *Agrobacterium*-mediated genetic transformation of tamarillo callus cells, aimed at establishing a transformed cell suspension culture, is yet to be described.

Callus cultures may be used for sustainable and large-scale production of secondary metabolites in pharmaceuticals, cosmetics, food, and related industries [7]. Mainly, transient gene expression accomplished through *Agrobacterium* transformation allows the production of high protein yields in a platform able to scale-up under aseptic conditions, where the culturing conditions can be more easily controlled [35]. Furthermore, since non-competent callus is unable to perform a regular differentiation path, it has the potential to establish itself as a stable platform for the expression of molecules of interest.

The selection of the explant and its physiological viability represents an essential initial step for the success of the transformation protocol [36]. Thus, actively growing sub-cultured callus masses were selected as explants for both transformation protocols tested.

In this study, two sequential approaches were performed to transform tamarillo callus cells. The first one followed the multiwell and vacuum filtration technique, which already described by Cordeiro et al. [21] for tamarillo embryogenic callus, in order to optimise the protocol for undifferentiated masses and check if there was an *Agrobacterium* strain that would confer a higher transformation efficiency of these cells.

The second approach aimed to optimise a method previously described by Chu et al. [22] by developing a straightforward liquid infection protocol tailored to tamarillo cell suspension cultures. This approach avoids vacuum incubation while still efficiently removing bacteria during the washing step, which, given the highly friable nature of the tamarillo callus cell lines [8], poses a challenge in achieving complete bacterial removal. Moreover, the success of this procedure was confirmed through an in vivo procedure, as reported by Chu et al. [22].

Analysing the results of the first transformation approach, it was found that inoculation intensity, which may be set to include, either individually or in various combinations, the cell density of the inoculum, the duration of inoculation and/or cocultivation, and the vacuum pressure and duration, plays a critical role in determining the obtained GUS response. Increasing the inoculation intensity either by increasing the cell density of the inoculum, by longer inoculation and/or cocultivation times, by higher vacuum pressure, or by prolonged vacuum infiltration increases the response obtained in the form of transient GUS expression, including the number of spots per explant [37]. As for the cell density, as recommended by Lopez et al. [38] and by Niedbała et al. [39] for other Solanaceae species, the culture densities used in this work ranged from 0.6 to 0.8.

Considering that using a liquid cocultivation medium and applying vacuum during this process results in a higher transformation frequency [40,41,42], in the first protocol described here, we used an inoculation time of 10 min under vacuum, followed by 3 days of incubation, which proved to be enough.

Concerning the composition of the inoculation solution, PVP and coconut water were added to the cocultivation to reduce the damage of explants by *Agrobacterium* and the phenolic production by the cells, as explained by Priya et al. [43], which indeed proved to prevent the browning of the cells. Acetosyringone and glutamine were also included in this solution since they enhance the transformation [44,45,46].

In our assays, it was observed that the proposed inoculum/bacteria ratio of 1 mg callus/2.5 mL, the time of vacuum infiltration, and cocultivation in a liquid medium produced a high GUS expression. Moreover, the insertion of the *gusA* gene in the callus cells and its expression are supported by the fact that the used binary vector p35SGUSINT presents an intron in the *gusA* coding sequence, which ensures that the gene is not expressed in the bacteria but only upon transfer to the plant [25].

For the bacterial removal (CEFO and CARB) and the selection of the transformed cells (KAN), a set of standard antibiotics well described in the literature was chosen [47,48,49]. The KAN concentration was increased gradually throughout the transfers of selected transformed cells, avoiding the negative impact of KAN on tissue growth, as demonstrated by Gerszberg and Grzegorczyk-Karolak [47] for *S*. *lycopersicum*. In the last transfer, since masses growth decelerated, KAN concentration was decreased again to enhance masses growth. As for the CEFO and CARB concentration, it decreased gradually in the last transfers, once again due to the deceleration of the masses growth, given that they may inhibit callus growth [50].

As for the amplification reactions performed to confirm the presence of the *gusA* gene in the genome of the cells used in the transformation procedure, the PCR amplification (Figure 2(C1)) and the qPCR quantification (Figure 2(C2)) confirmed that the transformation was successful. Also, no significant differences were observed in the transformation efficiency between using the LBA4404 *Agrobacterium* strain or the EHA105 one, proving that both strains are effective for tamarillo callus cells’ transformation. These findings are in line with what was reported for other Solanaceae species [49,51].

In the second approach, only the LBA4404 strain was used, as there were no significant differences in transformation success between the LBA4404 and EHA105 strains in the previous procedure. The vector used carried the *eyfp* gene, which expresses a fluorescent protein in the transformed plant cells but not in the bacteria, as the gene includes the IV2 plant intron.

As reported by Chu et al. [22], the introduction of a 5 min sonication step facilitated efficient transformation, likely by creating micro-wounds in the callus. Additionally, as noted by Ferreira et al. [52], the use of a liquid coculture eliminates the need for vacuum incubation to promote the adhesion of *Agrobacterium* to plant cells and subsequent gene transfer, a procedure that can be ineffective if not executed properly. With this optimised protocol, we mitigated this critical step; the coculture of plant cells with the bacterial suspension is sufficient for *Agrobacterium* gene transfer, resulting in a more streamlined and direct transformation protocol for tamarillo cell suspension cultures.

The washing steps described here, also utilising liquid media, allowed for efficient bacterial removal, which is typically challenging in liquid cultures, especially given the tissue’s friable and mucilaginous properties [8] that can entrap bacteria.

The incorporation of a fluorescent protein reporter gene enabled in vivo detection of transformed cells without compromising the callus masses used for transformation confirmation. Provided the analysed cells remain in a sterile environment, they can be selected based on fluorescence emission for future applications. The results obtained demonstrated a fluorescent pattern consistent with expectations, as described by Martínez Márquez et al. [53], with the *eyfp* protein localised around the nucleus, likely in the endoplasmic reticulum.

Nevertheless, for both approaches, some cells with lower or without GUS or EYFP expression were observed. Those masses still require continuous subsequent subculturing to effectively select the cells that present GUS or EYFP expression, enabling the establishment of transformed cell lines. Furthermore, optimising the combinations of key factors influencing inoculation intensity, such as inoculum cell density, inoculation duration, cocultivation conditions, as well as vacuum pressure and duration, could help increase the transformation efficiency, either by an increase of the percentage of transformed cells or by more stable levels of transgenes expression.

## 5. Conclusions

This optimised protocol presents an innovative transient transformation method for friable callus derived from the solanaceous tree tamarillo (*Solanum betaceum* Cav.), utilising *Agrobacterium tumefaciens*. This approach eliminates the need for prior transformation of leaves or other explants before callus induction, streamlining the process and potentially enhancing efficiency.

The screening method used to detect transformed tamarillo callus is straightforward: the reporter genes *gusA* and *eyfp* are integrated into the DNA regions transferred to the callus cells’ genome, allowing for visual selection of the transformed masses. Additionally, the transferred DNA regions include a kanamycin resistance gene, ensuring that only transformed cells survive in media containing this antibiotic, thereby streamlining the selection process. Moreover, through PCR and qPCR amplification, confirmation of the transient transformation and quantification of the transformation efficiency are possible.

Overall, as demonstrated by the results of the second approach, we present a simple method for the *Agrobacterium* transformation of tamarillo friable callus cells, confirming the insertion of a gene of interest through straightforward in vivo observation. Transformed callus was successfully obtained after three to five months.

This study represents a significant milestone in establishing tamarillo-transformed cell suspension cultures, particularly from dedifferentiated callus cells, as a reliable tool for future production of valuable molecules. The potential for scaling-up these cultures to bioreactors is already being tested, with promising preliminary results in terms of yield. This suggests advantages for industrial applications, positioning tamarillo cell cultures as a conceivable efficient platform for the large-scale production of molecules with proven interest. Transformation of tamarillo cell cultures with metabolite pathway genes and subsequent quantification of metabolites production could provide evidence to support the use of this protocol for the production of valuable molecules.

## Figures and Tables

**Figure 1 plants-14-01028-f001:**
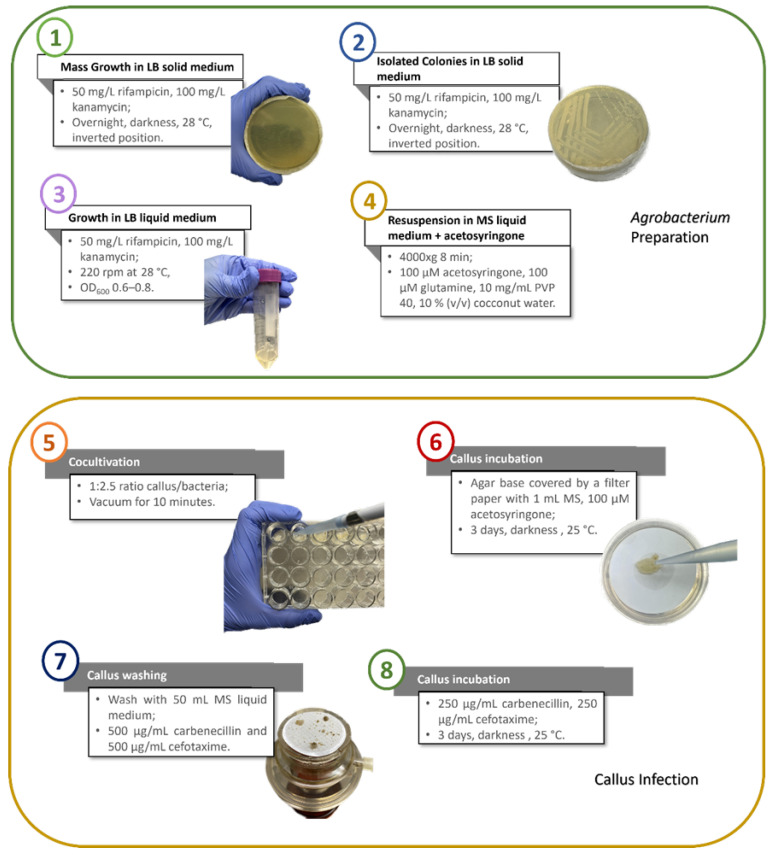
Steps of the optimized *Agrobacterium*-mediated transformation of tamarillo (*Solanum betaceum*) callus cells. The first step, *Agrobacterium* preparation (above: green), involves mass growth on LB solid medium (1), isolation of colonies on LB solid medium (2), growth in LB liquid culture to an OD_600_ of 0.6–0.8 (3), and resuspension in MS medium (4). The second step, callus cell infection (below: yellow), includes callus cocultivation with bacteria (5), incubation (6), washing (7), and re-incubation in the selective medium (8).

**Figure 2 plants-14-01028-f002:**
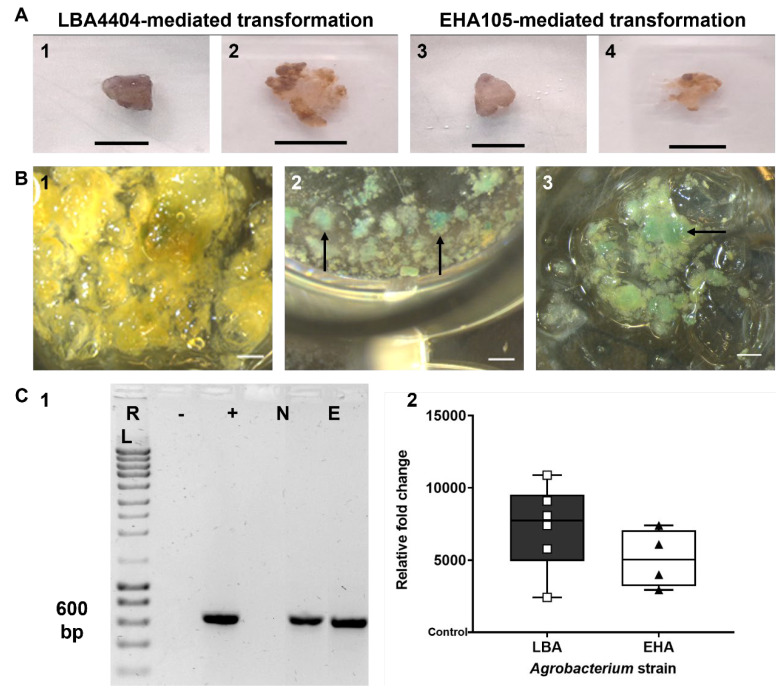
Results of the infection of tamarillo callus with *A*. *tumefaciens* containing the p35SGUSINT plasmid, through an infection in a multiwell plate. (**A**) Callus masses transformed with LBA4404 (**A1**,**A2**) and EHA105 (**A3**,**A4**) strains before selection (incubated in TP medium supplemented with 250 mg/L CEFO and 250 mg/L CARB) (**A1**,**A3**) and after 15 weeks of the selection process (incubated in TP medium supplemented with 120 mg/L KAN and 100 mg/L CARB) (**A2**,**A4**). Scale bar = 10 mm. (**B**) GUS activity in callus masses used in the transformation procedure. Blue staining indicates the expression of the *gusA* reporter gene. (**B1**), control sample; (**B2**,**B3**), transgenic line carrying p35SGUSINT construction. Arrows indicate areas of the callus showing GUS staining. Scale bar = 1 mm. (**C**) Confirmation and quantification of the callus masses transformation. (**C1**), electrophoresis of the PCR products performed to confirm the success of the transformation procedure; R, molecular weight size marker (NZYDNA Ladder III, NZYTech); −, negative control reaction (no DNA added); +, positive control reaction (p35SGUSINT as template DNA); N, negative control for the presence of the *gusA* gene in callus cells (DNA from a callus mass not used in the transformation procedure as template DNA); E and L, genomic DNA of tamarillo callus cell lines transformed with p35SGUSINT by EHA105 (E) and LBA4404 (L) *A*. *tumefaciens* strains. (**C2**), relative fold change of *gusA* insertion, as proposed by Livak and Schmittgen [32]. All experiments were carried out using at least four biological replicates, and all results are displayed as the averages of the replicates, along with their standard deviation values.

**Figure 3 plants-14-01028-f003:**
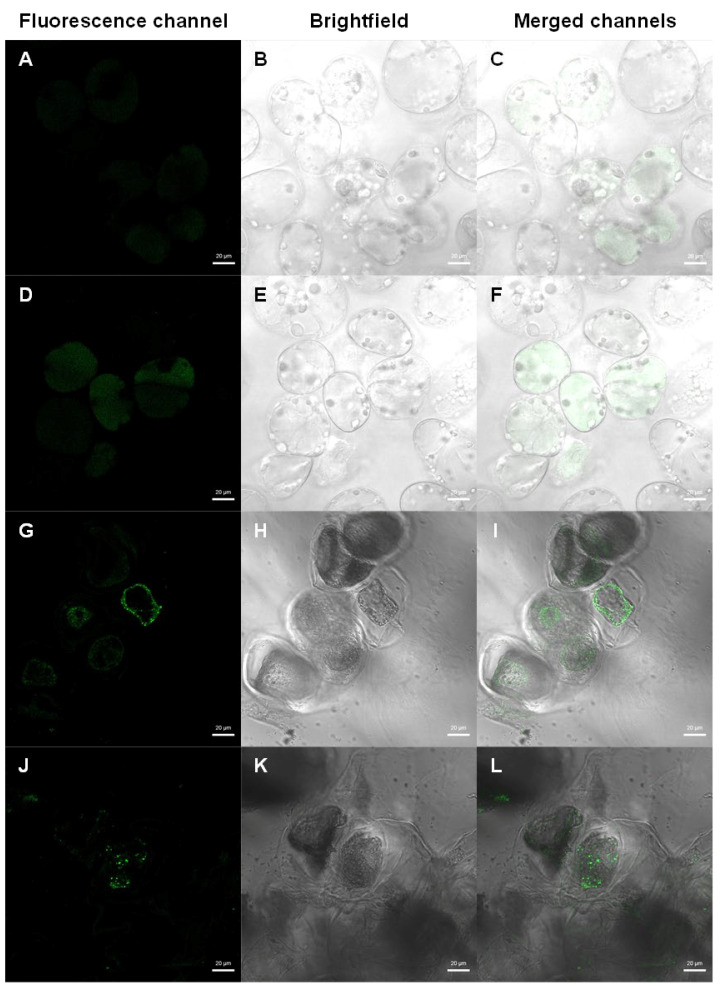
Fluorescence visualisation of tamarillo callus transformed in a liquid medium. Control samples (**A**–**F**) show a scattered and pale autofluorescence, unlike the transformed ones (**G**–**L**), which show the typical strong EYFP expression around the nucleus of the cells, probably in the endoplasmic reticulum. (**A**,**D**,**G**,**J**) Fluorescence channel; (**B**,**E**,**H**,**K**) brightfield; (**C**,**F**,**I**,**L**) merged channels. Scale bar = 20 µm.

**Table 1 plants-14-01028-t001:** Antibiotic supplementation on each cell transfer in the medium after the transformation procedures. After 18 weeks, the presented supplementation was maintained for 44 weeks. All the concentrations are in mg/L. CARB, carbenicillin; CEFO, cefotaxime; KAN, kanamycin.

Time	3 Weeks	6 Weeks	9 Weeks	12 Weeks	15 Weeks	18 Weeks	Subsequent Weeks
TP solid medium + antibiotics (transformation in the multiwell)	250 CARB	250 CARB	250 CARB	200 CARB	100 CARB	100 CARB	
	250 CEFO	250 CEFO	250 CEFO	200 CEFO	--	--	
	50 KAN	100 KAN	120 KAN	120 KAN	120 KAN	100 KAN	
TP solid medium + antibiotics (liquid medium transformation)	0 CARB	0 CARB	0 CARB	0 CARB	0 CARB	0 CARB	100 CARB
	300 CEFO	250 CEFO	250 CEFO	250 CEFO	250 CEFO	250 CEFO	--

## Data Availability

The original contributions presented in this study are included in the article. Further inquiries can be directed to the corresponding author.

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
