# Peer review of "Mediated Transformation of Tamarillo (Solanum betaceum) Callus Cell Suspension Cultures: A Novel Platform for Biotechnological Applications"

_plants, 2025, doi:10.3390/plants14071028_

Round 1
Reviewer 1 Report
Comments and Suggestions for Authors
This paper presents an Agrobacterium-mediated transient transformation protocol using tamarillo cell suspension cultures. The method is described in detail, the data provided in the manuscript can support the conclusions, and the article is well-written without any apparent flaws. Although this protocol represents a significant advancement in transient transformation of tamarillo callus, but there exits some negative cells or cells with low expressions of GUS or EYEP in Figure 2B2 and B3, Figure 3L. How can this issue be addressed?
Author Response
|
Comments 1: “Although this protocol represents a significant advancement in transient transformation of tamarillo callus, but there exits some negative cells or cells with low expressions of GUS or EYEP in Figure 2B2 and B3, Figure 3L. How can this issue be addressed?”
|
|
Response 1: Thank you for pointing this out. We agree with this comment. Therefore, we have added this paragraph to the end of discussion section, page 13, to address the issue referred. “Nevertheless, for both approaches, some cells with lower or without GUS or EYFP expression were observed. Those masses still require continuous subsequent subculturing to effectively select the cells that present GUS or EYFP expression, enabling the establishment of transformed cell lines. Furthermore, optimising the combinations of key factors influencing inoculation intensity, such as inoculum cell density, inoculation duration, cocultivation conditions, as well as vacuum pressure and duration could help increase the transformation efficiency, either by an increase of the percentage of transformed cells or by more stable levels of transgenes expression.” |

Reviewer 2 Report
Comments and Suggestions for Authors
My main concern is that this paper aims to establish an optimized method for Agrobacterium-mediated Tamarillo callus transformation. However, I did not see any data comparing different experimental conditions. For instance, authors claimed that PVP and coconut water prevented browning of cells after transformation. Please see in the conclusion: "PVP and coconut water were added to the cocultivation to reduce the damage of explants by Agrobacterium and the phenolic production by the cells, as explained by Priya et al. [45], which indeed proved to prevent the browning of the cells". However, I was unable to find evidence within the manuscript that supports this claim. The same issue with "The cocultivation with the bacterial suspension was made in a multiwell plate in a ratio of 1:2.5 (mg callus/μL bacteria suspension), that revealed to be the most successful procedure concerning the cell recovery". No actual data support this claim.
Some minor issues:
- Given the potential of tamarillo callus cell suspension cultures for producing valua-ble bioactive compounds, such as antioxidants and anti-inflammatory agents [21]. This reference is about cotton, not tamarillo. I would suggest authors to check citations and make sure all papers are correctly cited.
- Figure 2C2. I am not sure a box-plot is a good way to present this data if the data contains only two technical replicates. If the goal is to determine which strain is more efficient in transformation, why were multiple biological replicates not included?
Author Response
|
Comments 1: “My main concern is that this paper aims to establish an optimized method for Agrobacterium-mediated Tamarillo callus transformation. However, I did not see any data comparing different experimental conditions. For instance, authors claimed that PVP and coconut water prevented browning of cells after transformation. Please see in the conclusion: "PVP and coconut water were added to the cocultivation to reduce the damage of explants by Agrobacterium and the phenolic production by the cells, as explained by Priya et al. [45], which indeed proved to prevent the browning of the cells". However, I was unable to find evidence within the manuscript that supports this claim. The same issue with "The cocultivation with the bacterial suspension was made in a multiwell plate in a ratio of 1:2.5 (mg callus/μL bacteria suspension), that revealed to be the most successful procedure concerning the cell recovery". No actual data support this claim.”
|
|
Response 1: Thank you for pointing this out. Concerning prevention of browning, Figure 2A shows the transformed callus masses, which, after the selection procedure, present whitish areas, demonstrating, this way, that the protocol performed did indeed prevent the browning of the cells, as mentioned in section 3.2., in page 7 (“As Figure 2A shows, the callus cells at the final selection procedure showed limited growth and whitish areas (Figures 2A2, 2A4) compared with the beginning of the transformation procedure (Figures 2A1, 2A3), for both strains”). Additionally, as noted in the article, this step was carried out following the referenced studies (Priya et al. [45] and, in particular, Cordeiro et al. [23]), where these aspects were specifically discussed in relation to tamarillo embryogenic callus. As for the cocultivation procedure, we modified the referred sentence (first sentence of the third paragraph of section 3.1., in page 6) to clarify that the procedure was “successful” and not “the most successful”, given that, in fact, we did not try a different procedure. With “a successful procedure” we want to convey the idea that the use of multiwell plate for the cocultivation is a practical way of recovering the cells after the cocultivation in the wells. “The cocultivation with the bacterial suspension was made in a multiwell plate in a ratio of 1:2.5 (mg callus/µL bacteria suspension), that revealed to be a successful procedure concerning the cell recovery.”
|
|
Comments 2: “Some minor issues: Given the potential of tamarillo callus cell suspension cultures for producing valuable bioactive compounds, such as antioxidants and anti-inflammatory agents [21]. This reference is about cotton, not tamarillo. I would suggest authors to check citations and make sure all papers are correctly cited.”
|
|
Response 2: We agree with the suggestion, it was a lapse, so the reference was deleted. This statement near the end of the introduction is a conclusion of the above mentioned, so it does not demand a reference to support it.
|
|
Comments 3: “Figure 2C2. I am not sure a box-plot is a good way to present this data if the data contains only two technical replicates. If the goal is to determine which strain is more efficient in transformation, why were multiple biological replicates not included?”
|
|
Response 3: When we mention at the end of section 2.8. “The assay included non-template controls, and all reactions were run in two technical replicates” this refers to technical replicates for each qPCR reaction performed with the same genomic DNA sample. However, as mentioned in the first sentence of section 2.10., “all experiments were carried out using at least four biological replicates, and all results were displayed as the averages of the replicates, along with their standard deviation values”, therefore the use of a box-plot to present this data seems appropriate to us. In order to make this information clearer, we have also included it in the end of figure 2 caption (page 9). |

Reviewer 3 Report
Comments and Suggestions for Authors
Review Comments
The manuscript " Development of an Agrobacterium-mediated transient transformation protocol for potential metabolite production in Solanum betaceum Cav. callus " by Ferraz et al. is a good study about transient transformation protocol on Solanum betaceum. This research article provides importance of transient assay in crop plants. This manuscript is of interest to the crop biology researchers, as well as different biology researchers and I expect that the article will be well-cited, but it lacks proper validation of this protocol. I have the following major comments to consider.
- Authors provide high resolution figures and graphs. Graphs should have data points.
- Methods and material about florescence imaging lacks all detail about confocal imaging like gain, magnification, etc. Along with authors claimed about EYFP around nucleus and ER without any colocalization with marker genes.
- Only expression of marker genes don’t make any conclusion about metabolite productions. Authors should express metabolite pathway genes and quantify the desired metabolite from the callus.
Author Response
|
Comments 1: “Authors provide high resolution figures and graphs. Graphs should have data points.”
|
|
Response 1: Thank you for pointing this out. We modified Figure 2C2 (page 9). |
|
Comments 2: “Methods and material about florescence imaging lacks all detail about confocal imaging like gain, magnification, etc. Along with authors claimed about EYFP around nucleus and ER without any colocalization with marker genes.” |
|
Response 2: We agree with the suggestion. We have, accordingly, modified section 2.9. to address this issue. “When the callus transformed in the liquid medium revealed yellowish colour masses with no evidence of cell necrosis, i.e. appearance of transformed masses, transformed callus and control samples were observed under a Zeiss LSM 710 (Carl Zeiss AG, Oberkochen, Germany) confocal microscope with magnification of 40× and 0.99 AU of pinhole diameter. EYFP fluorescence was viewed by excitation with a 488 nm Argon laser, a detection wavelength of 516-617 nm and a detector gain of 627 for the EYFP channel and 500 for bright-field. Brightness and contrast were adjusted using the Zeiss Zen 2 (Blue Edition, Carl Zeiss AG) software.” As for the localisation of EYFP expression, no marker genes were used, because the location of the nucleus in the cell type is quite visible in the bright-field. |
|
Comments 3: “Only expression of marker genes don’t make any conclusion about metabolite productions. Authors should express metabolite pathway genes and quantify the desired metabolite from the callus.” |
|
Response 3: The comment is pertinent. However, given the context provided in the introduction concerning the production of metabolites in plant cell cultures, the main objective of this work was to develop an optimised tool for that purpose. With the results presented in this article, we believe that, although the procedure you mention could provide more evidences about metabolite production, this protocol already has the potential to be a tool for this purpose as referred in the conclusion (“This study represents a significant milestone in establishing tamarillo-transformed cell suspension cultures, particularly from dedifferentiated callus cells, as a reliable tool for producing valuable molecules. The potential for scaling up these cultures to bioreactors is already being tested, with promising preliminary results in terms of yield. This suggests clear advantages for industrial applications, positioning tamarillo cell cultures as an efficient platform for the large-scale production of molecules with proven interest.”). Thus, in the end of conclusion section (page 13), a sentence referring the need for this additional step was added “Transformation of tamarillo cell cultures with metabolite pathway genes and subsequent quantification of metabolites production could provide evidence to support the use of this protocol for the production of valuable molecules.” |

Round 2
Reviewer 2 Report
Comments and Suggestions for Authors
Thanks for addressing the comments.
Author Response
Comments: Thanks for addressing the comments.
Response: [The comments were adressed in the previous manuscript version].
Reviewer 3 Report
Comments and Suggestions for Authors
Authors didn’t provide any evidence of metabolite production, so I will suggest the authors remove all the metabolite production terms from title, abstract, etc. and present or rewrite this paper as only transient gene expression system of Solanum betaceum.
Comments on the Quality of English LanguageIt is good
Author Response
|
1. Summary |
|
|
|
Thank you very much for taking the time to review this manuscript. Please find detailed responses below and the corresponding revisions and corrections in track changes in the re-submitted files.
|
||
|
2. Point-by-point response to Comments and Suggestions for Authors |
||
|
Comments 1: “Authors didn’t provide any evidence of metabolite production, so I will suggest the authors remove all the metabolite production terms from title, abstract, etc. and present or rewrite this paper as only transient gene expression system of Solanum betaceum.”
|
||
|
Response 1: Thank you for the recommendation. Sentences along the manuscript were modified to focus the aim of the work on tamarillo transient transformation. These modifications were made in the following sections: Title (page 1); First sentence of the Abstract (page 1); Fifth to seventh sentences of the second paragraph of the Introduction (page 2); Last sentence of the fourth paragraph of the Introduction (page 2); Last paragraph of the Introduction (page 2); First sentence and last two sentences of the last paragraph of the Conclusions section (page 13). The only references to metabolite production kept were the ones included to contextualise tamarillo suspension cultures as a system with potential for metabolite production, a possible future application that needs further confirmation, even though not related with the objective of this work. This way, the following sections were kept: Last sentence of the Abstract (page 1); First to fourth and eighth to ninth sentences of the second paragraph of the Introduction (page 2); Two first paragraphs of the Discussion section (page 10). |
||

Round 3
Reviewer 3 Report
Comments and Suggestions for Authors
Authors have responded all my comments. No more comments.
Comments on the Quality of English LanguageGood